# Surface Roughness of Polyetheretherketone Printed by Fused Deposition Modeling: A Pilot Study Investigating the Impact of Print Layer Thickness and Polishing Method

Tânia Soares [1], Carlos Fernandes [2,3,*], Cláudia Barbosa [4,5], Mário A. P. Vaz [2,3], Tiago Reis [4,5] and Maria Helena Figueiral [1,2]

1  Faculty of Dental Medicine, University of Porto, 4200-393 Porto, Portugal; tania_soares89@hotmail.com (T.S.); hfigueiral@gmail.com (M.H.F.)
2  Institute of Science and Innovation in Mechanical Engineering and Industrial Engineering (INEGI), 4200-465 Porto, Portugal; gmavaz@fe.up.pt
3  Faculty of Engineering, University of Porto, 4200-465 Porto, Portugal
4  Faculty of Health Sciences, University Fernando Pessoa, 4249-004 Porto, Portugal; cbarbosa@ufp.edu.pt (C.B.); tiagofaria@ufp.edu.pt (T.R.)
5  FP-I3ID, Faculty of Health Sciences, University Fernando Pessoa, 4249-004 Porto, Portugal
*  Correspondence: cfernandes@fe.up.pt

**Abstract:** Polyetheretherketone is a high-performance thermoplastic polymer that can be used in 3D printing by fused deposition modeling, and is a promising material for dental applications. Some printing parameters are sensitive and can influence the properties of the printed object. Thus, this work aims to evaluate the influence of the print layer thickness on the surface roughness of polyetheretherketone before and after polishing and to verify the effectiveness of the polishing method used, as well as to compare it with the results obtained using polymethyl methacrylate as the control group. Specimens with different impression layer thicknesses were printed with polyetheretherketone (Group A—0.1 mm and Group B—0.3 mm). Additionally, a control group with polymethyl methacrylate specimens was milled. Roughness evaluation was conducted using a contact profilometer after the specimens had been printed (before polishing). Then, silicon carbide sandpaper was used to polish the surface, and the roughness was reassessed. Differences were observed between specimens regarding the print layer thickness and the roughness, with the 0.3 mm layer thickness showing the lowest roughness values. The results of this pilot study suggest that the surface roughness of fused deposition modeling printed polyetheretherketone is influenced by print layer thickness, with the lowest roughness seen at a thickness of 0.3 mm.

**Keywords:** three-dimensional printing; polyetheretherketone (PEEK); fused deposition modeling (FDM); roughness; print layer thickness; silicon carbide polishing

## 1. Introduction

Polyetheretherketone (PEEK) is a semi-crystalline high-performance thermoplastic polymer developed by Imperial Chemical Industries in 1977. Due to its interesting mechanical and thermal properties, like modulus of elasticity, mechanical resistance, rigidity, and lightness, it has been considered an alternative to metals such as titanium and zirconium. Because this material has a modulus of elasticity and ultimate tensile stress similar to human bone, enamel, and dentin, it is suitable for use in dentistry, particularly in oral rehabilitation, for dental implants, removable prostheses, crowns, and fixed bridges [1–7]. Its glass transition point is approximately 143 °C, and its melting point is 343 °C [1,6,8–10]. Moreover, it allows good polishing, which promotes less bacterial plaque adhesion, and presents good wear resistance [2,11–14].

Initially, PEEK was processed using traditional casting methods, which were time-consuming and unsuitable for manufacturing small items. Later, through computer-aided

design/computer-aided manufacturing (CAD/CAM) technology, PEEK disks were used to mill crowns, fixed partial dentures, removable prostheses, partial denture frameworks, obturating prostheses, and dental implants. However, low production rates, the excessive waste of grinding material, and an inability to produce more complicated and controlled architectures are the disadvantages of subtractive technology [15,16].

Fused deposition modeling (FDM) is a method in which layers of molten material are deposited from a filamentous nozzle and then solidified within 0.1 s [17]. This printing technique has become increasingly important, as it allows the construction of parts with complex geometries and is a more economical and simpler method due to using filament instead of powder [18]. Some authors refer to FDM as an alternative method to manufacture complex PEEK structures due to it being a low-cost, easy, and minimal-waste 3D printing technology. This technique overcomes some CAD/CAM milling problems and generates items more resistant to aging since PEEK filaments absorb less moisture than PEEK milling blocks [16]. Moreover, 3D printing in PEEK is an innovation in dentistry that allows for greater effectiveness and efficiency and is a quick and inexpensive process [9,19].

Several parameters can modify the outcome of FDM printing, influencing the mechanical behavior and properties of the devices, such as print layer thickness, print wire width, printing speed, extruder temperature and diameter, filament feed rate, viscosity, filament quality, and the printing routine itself. Therefore, initial tests are necessary to understand the ideal parameters for each filament [16,20–22]. The influence of FDM printing parameters on the printed PEEK parts' dimensional stability and mechanical properties is attracting increasing interest in research. Moby et al. [16] concluded that the surface roughness of 3D FDM-printed PEEK seems suitable for dental restorations. However, information about the surface characteristics of FDM PEEK remains unclear [16,20–22].

Given the direct correlation between surface roughness and biofilm formation, polishing materials designed for use in the oral cavity becomes crucial. This step is essential for aesthetic reasons and also plays a significant role in minimizing bacterial plaque accumulation and enhancing resistance to the fluids present in the oral environment. A rough surface tends to suffer discoloration or pigmentation, increases bacterial accumulation, and may promote abrasion wear of opposing dental elements. Therefore, it is extremely important to evaluate the surface topography, roughness, hardness, and abrasion of materials that may be used in the oral cavity [23,24]. Kurahashi et al. [23] stated that humans can detect roughness values of at least 0.5 μm.

According to different authors, the print layer thickness is decisive for the printed parts' surface roughness and dimensional accuracy [10,11,21,25]. Due to the layer-by-layer deposition printing process, the surfaces of objects made by 3D printing have inevitable irregularities.

Optimizing printing parameters to enhance the surface roughness of printed objects may lead to an extended printing time. Thus, the alternative is often to finish the surface after printing using techniques such as mechanical polishing [8,25,26]. This is a common surface processing method in dentistry to obtain a smooth surface. Heimer et al. [24] tested different polishing techniques on PEEK and polymethyl methacrylate (PMMA) samples and concluded that different material specimens were similarly polishable. It is worth mentioning that PMMA is a material widely used in the oral cavity and the gold standard in the manufacture of dentures and occlusal splints due to its mechanical proprieties, easy processing, repair techniques, and good polishing ability [5,27,28]. Its use as pre-formed blocks for milling is associated with lower porosity [29]. However, there is no consensus among authors on the best technique for polishing FDM PEEK or milled PMMA [21,30–32].

This study established two objectives. First, we aim to evaluate how the print layer thickness affects the surface roughness of FDM-printed PEEK before and after mechanical polishing. Additionally, we seek to assess the effectiveness of polishing PEEK with SiC sandpaper and compare it to PMMA polished with the same technique.

## 2. Materials and Methods

This research used a PEEK filament (Peek Ketaspire KT-820 NT Filament, Medphen, Madrid, Spain) for Class I Medical Devices (ISO 10993-1) [33]. Following the manufacturer's instructions, the filament was dried in an oven at 150 °C (Vismec DW25, Moretec, Hoorn, The Netherlands) for 4 h before the printing process and after printing to prevent water absorption. An adhesive (Nano Polymer Adhesive, Vision Miner, Irvine, CA, USA) was applied on the surface of the print table to promote the adhesion of the specimens to the table and prevent the printed pieces from warping. Printing was performed on the FDM 3D printer AON-M2 (AON3D, Montreal, QC, Canada), on which initial tests were carried out to assess the technical parameters (printing path, printing speed, print layer thickness, extruder temperature), as recommended by the manufacturer. The final specimens and the stabilization gutter prototype were printed with the printer-adjusted parameters summarized in Table 1. The selected printing path is shown in Figure 1.

**Table 1.** Printing parameters used.

| Parameters | Technical Specifications |
|---|---|
| Extruder diameter | 0.6 mm |
| Printing layer thickness | 0.1 mm or 0.3 mm |
| Print speed | 10 mm/s |
| Printing table temperature | 160 °C |
| Chamber temperature | 100 °C |
| Extruder temperature | 380 °C |

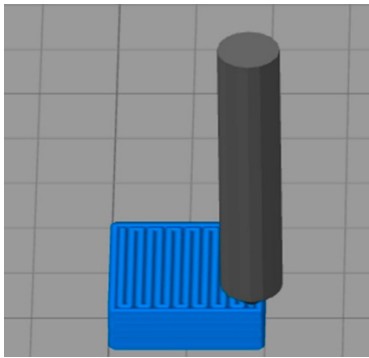

**Figure 1.** Printing path.

The specimens (10 mm × 10 mm × 1.5 mm) were designed using the software Solidworks (Version 2021) (Dassault Systèmes Solidworks Corporation, Waltham, MA, USA). Four specimens were printed—two specimens (A1 and A2) with 0.1 mm and two specimens (B1 and B2) with 0.3 mm of print layer thickness—to evaluate whether changing this parameter modifies the specimen's surface roughness. The surface roughness of all specimens was assessed after the printing process without any surface finishing. Subsequently, these specimens were polished with a series of silicon carbide (SiC) sandpapers of increasing grain (P80, P180, P320, P800, P1200, and P4000, Struers, Ballerup, Denmark) [18,34–36]. For this procedure, the specimens were first placed in a support material (Polyvinyl Siloxane, Aquasil Ultra, Dentsply Sirona, Charlotte, NC, USA) (Figure 2) to allow safe and uniform polishing. The polishers (Rotopol-21, Struers, Denmark) were used with tap water lubrication at 300 rotations per minute (rpm) for 60 s (Figure 3). After this procedure, the specimens' roughness was measured again.

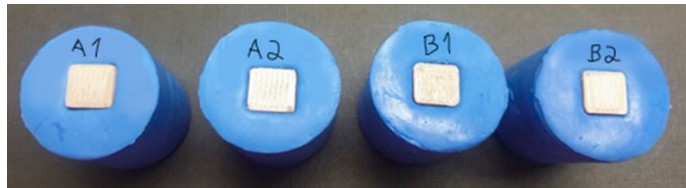

**Figure 2.** Packing the specimens in the support to perform their polishing.

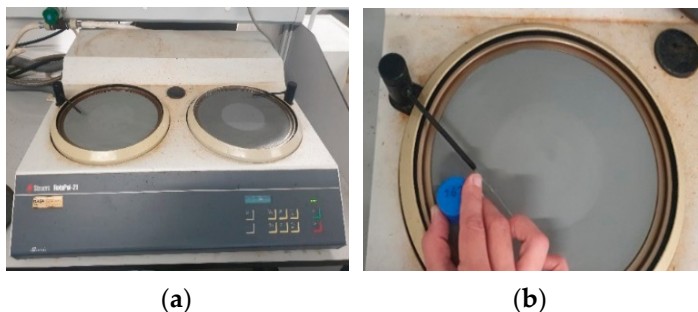

|     |     |
| :-: | :-: |
| (**a**) | (**b**) |

**Figure 3.** (**a**) Polishers (Rotopol-21, Struers, Denmark). (**b**) Polishing of a specimen.

The control group for comparison consisted of 3 specimens produced in PMMA (Aidite Temp, Aidite Technology Co, Qinhuangdao, China). Another STL file was created using computer software (Exocad DentalCAD, Exocad GmbH, Darmstadt, Germany) for specimens with the same dimensions as the PEEK specimens ($10 \times 10 \times 1.5$ mm). These PMMA specimens were then milled (DWX-52D, Roland DGA Corporation, Irvine, CA, USA). After milling, different surface treatments were carried out on these specimens: one specimen was left unfinished (P1), another was polished with the SiC sandpaper used for the PEEK specimens (P2), and another was subjected to a conventional finish with polishing drills and diamond paste (P3). Then, their roughness was evaluated.

A Hommelwerke T8000 controller with an LV-50 linear unit (Hommelwerke GmbH, Villingen-Schwenningen, Germany) was employed to obtain the specimens' surface topography (see Figure 4a). A TKL 300 pickup was used as the stylus probe, with a vertical measurement range of $\pm 300$ μm, a tip radius of 5 μm, and a cone angle of 90 degrees. A support structure was created in polylactic acid (PLA) using the FDM 3D printer to fix the specimens in place securely, ensuring accurate and repeatable readings from the stylus profilometer (see Figure 4b). A 7.5 mm $\times$ 3 mm evaluation area was defined on each specimen with a resolution of 1 μm $\times$ 10 μm. The 7.5 mm dimension was perpendicular to the printing direction, where higher roughness is expected. The topographies were subjected to a form removal (plane tilt) and a cut-off filter of 0.8 mm. The specimens' areal surface roughness parameters were determined per ISO 25178-2 to calculate the arithmetic mean of the surface height (Sa). Also, a profile was extracted perpendicularly to the printing direction, and the arithmetic mean roughness (Ra) was calculated according to ISO 4287, as well as the maximum height from the highest peak and the deepest valley (St). Data analysis was performed using Microsoft Excel (Version 2402) software (Microsoft, Microsoft Corporation, Redmond, WA, USA).

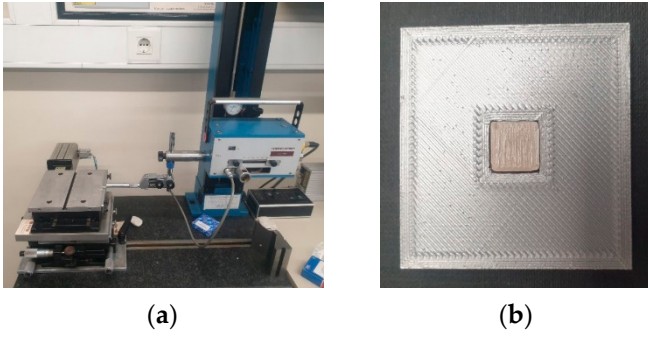

|     |     |
| :-: | :-: |
| (**a**) | (**b**) |

**Figure 4.** (**a**) Stylus probe profilometer (Hommelwerke GmbH, Germany). (**b**) Support for fixing the specimens for the surface topography reading.

## 3. Results

The surface topography of the two specimens (A1 and A2) with 0.1 mm and the two specimens (B1 and B2) with 0.3 mm of print layer thickness was assessed (1) after printing without any surface finishing (before polishing assessment) and (2) after polishing with SiC sandpaper (Figure 5).

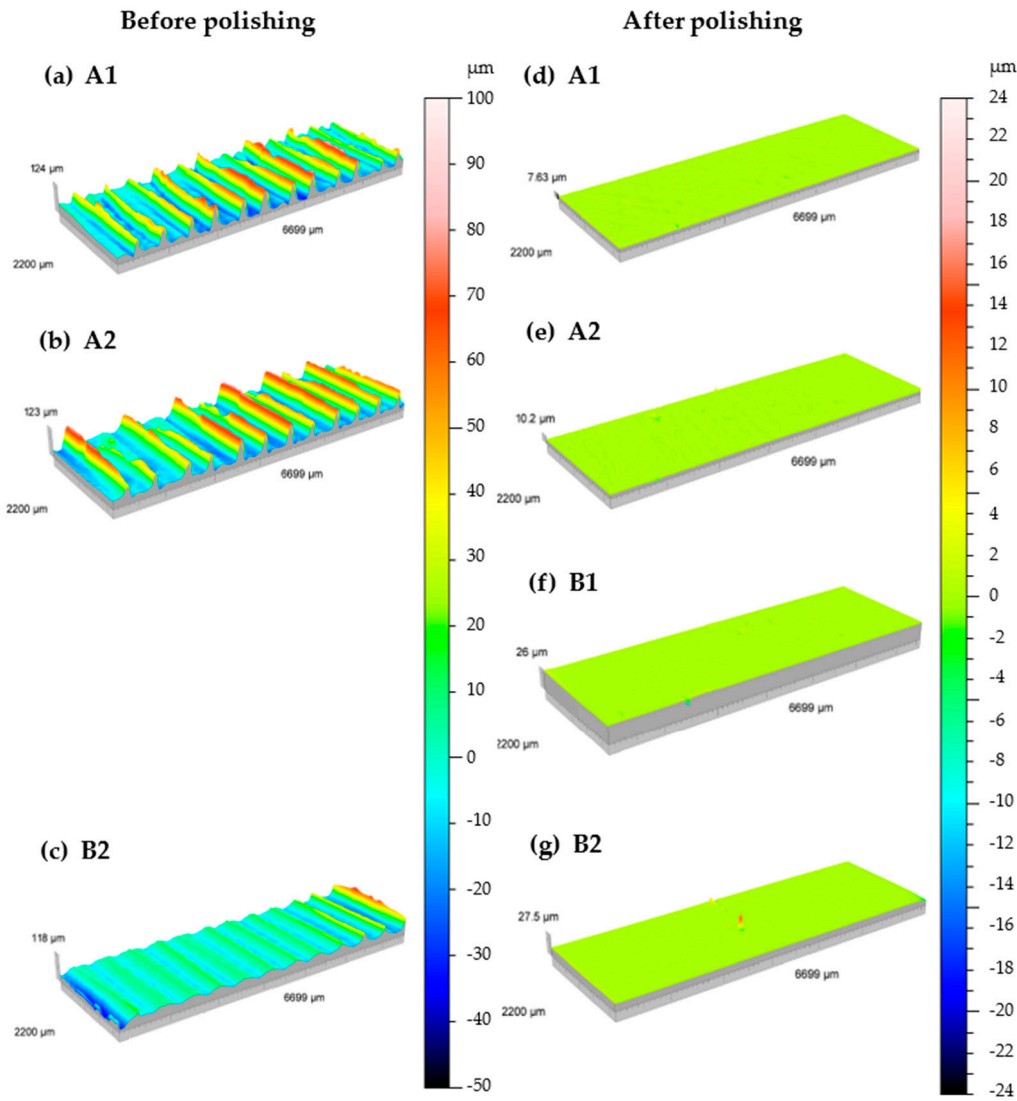

**Figure 5.** Surface topography of the specimens without any surface finishing (**a–c**) and after polishing with SiC sandpaper (**d–g**).

In the first analysis, specimen B1's roughness was higher than the profilometer tip's measurement range (approximately 300 µm), making it impossible to measure. A considerable difference was observed in the surface roughness of the specimens in the two evaluation moments, and, as expected, all specimens showed higher surface roughness in the first measurement (Table 2). After the polishing procedures, Ra, Sa, and St decreased significantly in each specimen.

**Table 2.** Roughness parameters before and after polishing for each specimen.

| Specimens | Roughness Parameters | Before Polishing (µm) | After Polishing (µm) |
|---|---|---|---|
| A1 | Sa | 17.3 | 0.19 |
|  | Ra | 0.91 | 0.12 |
|  | St | 124 | 7.64 |
| A2 | Sa | 18 | 0.15 |
|  | Ra | 1.05 | 0.09 |
|  | St | 123 | 10.2 |
| B1 | Sa |  | 0.09 |
|  | Ra |  | 0.05 |
|  | St |  | 26 |
| B2 | Sa | 6.51 | 0.1 |
|  | Ra | 0.33 | 0.05 |
|  | St | 118 | 27.5 |

Regarding the comparison between specimens with the same print layer thickness, it was not possible to compare B1 and B2 before polishing. However, all other comparisons (A1 with A2, before and after polishing, and B1 with B2 after polishing) showed very close values in the evaluated parameters, both before and after final polishing (Table 2).

Because the specimens differed in their printing characteristics—specifically, the print layer thickness—they were grouped according to this characteristic to calculate average roughness parameters (Table 3). Comparing the average values of Ra, Sa, and St of the two groups at each of the moments of roughness evaluation showed that the groups tended to differ. Group A always had a rougher surface than Group B (0.98 µm and 0.33 µm, respectively, before polishing), although this difference tended to be smaller after polishing (0.11 µm and 0.04 µm, respectively) (see Figure 6 and Table 3).

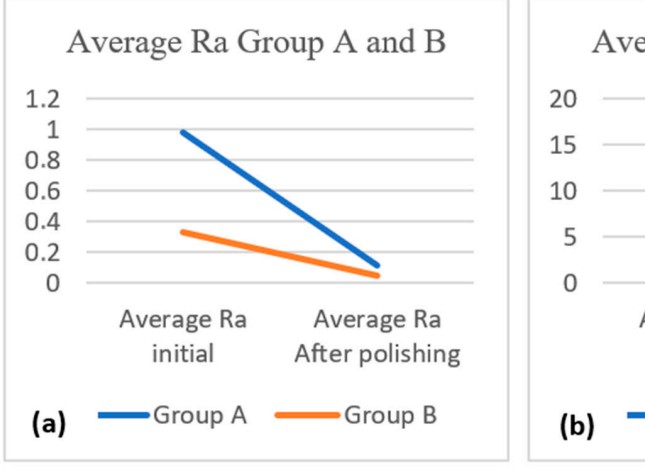 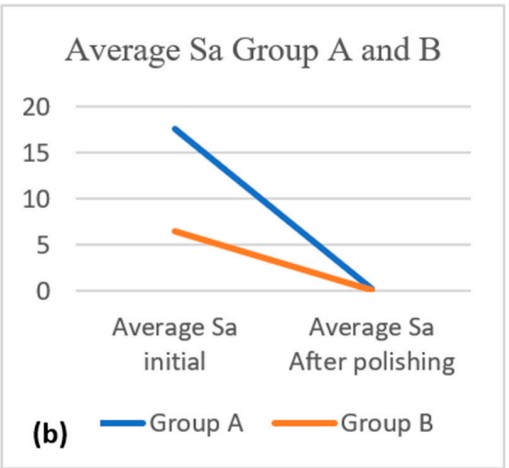

**Figure 6.** Difference between average Ra (**a**) and average Sa (**b**) of Groups A and B.

**Table 3.** Averages (Ra and Sa) before and after polishing for Groups A and B.

| Group | Group Average Ra (μm) | | Group Average Sa (μm) | |
|---|---|---|---|---|
| | Before polishing | After polishing | Before polishing | After polishing |
| A (A1 and A2) | 0.98 | 0.11 | 17.65 | 0.17 |
| B (B1 and B2) | 0.33 | 0.05 | 6.51 | 0.09 |

Regarding the PMMA specimens (P1, P2, and P3), surface topography was carried out on all specimens (Figure 7), and the parameters Sa, Ra, and St were assessed. A considerable difference was observed in the surface roughness between the unpolished and polished specimens. However, the specimens' surface roughness was similar in the different polishing steps applied, as shown in Table 4.

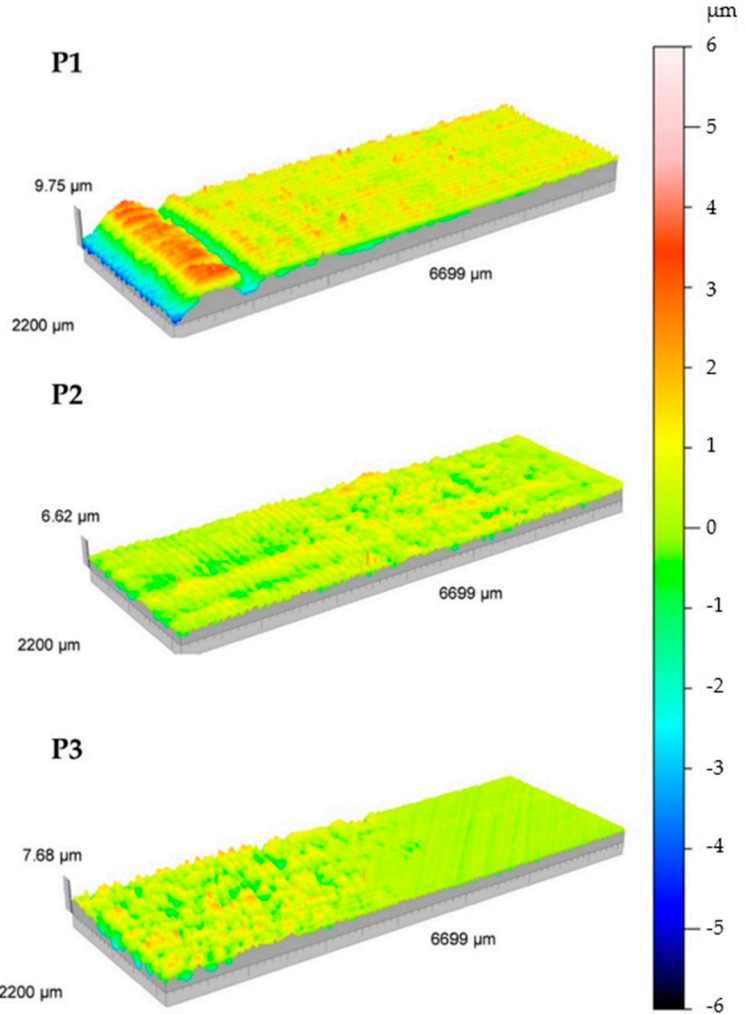

**Figure 7.** Three-dimensional image of the surface roughness of PMMA specimens P1, P2, and P3.

**Table 4.** Roughness test results for each PMMA specimen—Ra, Sa, and St.

| Specimens | | P1 (Unfinished Specimen) | P2 Polished with SiC Sandpaper | P3 (Conventional Finish) |
|---|---|---|---|---|
| Roughness Parameters | Sa | 0.74 | 0.4 | 0.49 |
| | Ra | 0.26 | 0.11 | 0.11 |
| | St | 9.75 | 7.68 | 6.62 |

When comparing the PEEK specimens with the PMMA specimens, although the PEEK specimens had higher roughness before polishing, after polishing with SiC sandpaper, the PEEK specimens had equal or lower roughness than the PMMA specimen under the same conditions (Table 5).

**Table 5.** Arithmetic mean roughness (Ra) before and after polishing PEEK and PMMA specimens.

| Specimens | Arithmetic Mean Roughness (Ra) before Polishing $\pm$ SD ($\mu$m) | Arithmetic Mean Roughness (Ra) after Polishing $\pm$ SD ($\mu$m) |
|---|---|---|
| PEEK—Group A | $0.98 \pm 0.07$ | $0.11 \pm 0.02$ |
| PEEK—Group B | $0.33 \pm$ * | $0.04 \pm 0$ |
| PMMA | $0.26 \pm$ * | $0.11 \pm$ * |

SD—standard deviation; * Not possible to calculate standard deviation because there is only one specimen.

## 4. Discussion

The aims of this study were (1) to determine how print layer thickness affected the surface roughness of FDM PEEK before and after mechanical polishing; and (2) to assess the effectiveness of polishing PEEK with SiC sandpaper and compare it to polishing PMMA with the same technique. This study demonstrated that printing PEEK specimens with a 0.3 mm print layer thickness produced inferior surface roughness both before and after SiC sandpaper polishing. When compared to the gold standard (PMMA), after polishing, the PEEK specimens had the same roughness in Group A (0.1 mm impression layer thickness) and lower roughness in Group B (0.3 mm impression layer thickness).

The advantages of 3D FDM printing include minimal time to produce items, a limited amount of material, and low cost. However, it has two disadvantages: (1) the need for postprocessing treatment to remove the support structures and polish the surface, and (2) a variable behavior because the printing conditions, such as nozzle temperature, chamber temperature, printing speed, and print layer thickness, directly affect the characteristics of printed objects [19,37]. Limaye et al. [22] reported that PEEK printing is technically complex compared to other thermoplastic materials melted at low temperatures. These authors performed a scanning electron microscope analysis of FDM-engineered PEEK samples and found irregularities created by material dragging by the nozzle during print. Optimizing printing parameters is crucial for controlling the mechanical properties of printed PEEK parts [22].

According to Wu et al. [14] and Li et al. [16], the print layer thickness is the parameter with the greatest influence on some mechanical properties of the printed structure. They claim that increasing the print layer thickness reduces the printed part's geometry precision and contour perfection and showed optimal mechanical properties in PEEK in samples produced with a layer thickness of 0.3 mm [14]. Some authors recommend a print layer thickness of 0.1 mm to reduce internal defects and improve the surface finish [21]. The print layer thickness seems to be a printing parameter that significantly affects the structure roughness. According to various authors, these two properties are directly related, because a decrease in print layer thickness generally causes a decrease in surface roughness [8,21,25,38]. The authors attribute these surface roughness results to the interlayer gaps caused by the FDM printing [38]. The results of the present study contradict those of the authors mentioned above [8,21,25,38] since the specimens with the highest roughness had the lowest print layer thickness (0.1 mm). Nonetheless, as mentioned previously, other printing conditions, such as nozzle temperature, chamber temperature, and printing speed, directly affect the characteristics of printed objects and may explain the difference between the present study and previous ones. Another hypothesis that could be raised is the utilization of multiple brands of PEEK across these studies and on different printers. This variation in materials and printing methods may have influenced the behavior observed, potentially contributing to the disparity in the results. The printing conditions should be described and protocolized to enable a comparison with future studies' results.

Defining an optimal polishing technique for PEEK is important because, regardless of the print layer thickness, mechanical finishing decreases surface roughness and can promote smooth and soft surfaces.

The maximum roughness of hard surfaces within the oral cavity after polishing, as indicated by several authors [23,24,39–42], should not exceed 2 μm, which represents the threshold for bacterial plaque accumulation. Nevertheless, alternative studies suggest that a clinically acceptable limit could be set at 10 μm [40,43].

Some authors state that assessing roughness using a contact-type roughness measuring device, such as the one used in this work, is not ideal due to the risk of damaging the sample surface [21]. However, Elawadly et al. [44] chose to evaluate roughness (Ra) using this method because the color of the PEEK samples causes inadequate light reflection, making the use of optical interferometry difficult. These authors considered this method effective due to promoting direct contact between the measuring tip and the sample surface. In the present study, the contact-type roughness measuring device was also used because the same difficulty was encountered when reading PEEK using optical interferometry. Other authors employ different approaches because there is no universal method for treating PEEK's surface.

Several investigations report different values when polishing milled PEEK with SiC sandpaper, namely, Keul et al. [45] (P500 to P2400) found Ra = 0.04 μm, Çulhaoglu et al. [45] (P1200) Ra = 0.53 μm, and Caglar et al. [45] (P600 and P800) Ra = 1.11 μm. A direct comparison cannot be made since no publication was found describing the same sequence of SiC sandpapers used in this work (P80, P180, P320, P800, P1200, and P4000) for FDM PEEK. Even so, given that lower roughness values were obtained with the method used (Ra between 0.04 μm and 0.11 μm), it is suggested that polishing FDM PEEK with SiC sandpaper of different grain sizes could be an effective technique.

Han et al. [18] conducted a study where the surface morphology of printed PEEK samples was determined. The authors evaluated both samples with three different surface treatments: (1) no polishing, (2) mechanical polishing (SiC sandpaper), and (3) sandblasting (120 μm alumina). As expected, the untreated specimens had the roughest surfaces (Sa = 17.67 μm), and the polished specimens had the smoothest surfaces (Sa = 0.42 μm). The same trend was observed in the present study, as the samples with and without polishing differed in surface roughness. However, although the initial roughness values of Group A agree with the results of Han et al. [18] (Sa before polishing in Group A = 17.65 μm), the initial roughness of the Group B samples was much lower (Sa before polishing in Group B = 6.51 μm). After polishing, both groups had lower roughness than that reported by those authors (final Sa of Group A = 0.17 μm and final Sa of Group B = 0.09 μm). Thus, the polishing technique used in the present work seems to be more effective, since the specimens from Group A, which had an initial roughness close to that described by Han et al. [18], showed a lower roughness after polishing than that reported by those authors.

The same was verified in the work of Gao et al. [21], which also analyzed the surface roughness of FDM PEEK specimens. They obtained Ra values of 0.61 to 0.66 μm before polishing and Ra values of 0.10 to 0.15 μm after polishing the specimens. The present study obtained lower roughness results after polishing (Ra = 0.04 and 0.11 μm), probably due to using a different polishing technique, although this cannot be confirmed as Gao et al. [21] did not mention the polishing protocol used. Thus, it can be assumed that the polishing method employed, involving P80, P180, P320, P800, P1200, and P4000 SiC sandpapers, seems to be suitable for rendering FDM PEEK clinically usable.

Kurahashi et al. [23] state that humans can detect roughness values of at least 0.5 μm. In fact, in this work, the specimens' roughness was within that limit (Ra = 0.04 μm and 0.12 μm). In addition, as already mentioned, the roughness limit for plaque accumulation in restorations inserted in the oral cavity should not be greater than 2 μm [23,24,31,40,42,45–47], so since the results obtained were lower than this value, there was no need to add another polishing method to the specimens' surface.

The roughness analysis of the PMMA specimens was aimed at comparison with the PEEK specimens, as PMMA is a gold-standard material in prosthodontics. PMMA had lower roughness values than PEEK for unpolished specimens (PMMA Ra = 0.26 μm; PEEK Ra = 0.33 μm and 0.98 μm), but the same Ra as PEEK specimens with a 0.1 mm print layer thickness after polishing with SiC sandpaper (PMMA Ra = 0.11 μm; PEEK Ra = 0.11 μm). The lowest roughness was found in PEEK specimens with a 0.3 mm impression layer after polishing with sandpaper (Ra = 0.04 μm). Therefore, although PEEK after FDM printing has a rough surface, finishing it with SiC sandpaper reduces roughness to values close to, or even lower than, the roughness values of the gold-standard material and at a clinically acceptable level for use in the oral cavity (Ra < 0.2 μm). However, the studies that report techniques for polishing milled PMMA [31,32] with SiC sandpaper (until P1200) present an Ra between 0.19 μm–0.35 μm, corroborating the idea that the polishing method used in this study is also clinically suitable for milled PMMA.

Since the present study is a pilot study, the sample size and the flat geometry of the printed parts are its main limitations. Future studies with larger sample sizes are needed to corroborate the present findings. Also, different geometries, such as removable prostheses, fixed provisional prostheses, and occlusal splints, should be tested to simulate clinical situations and demonstrate whether the polishing method is effective in these geometries.

## 5. Conclusions

The results suggest a relationship between the print layer thickness and the surface roughness of the FDM PEEK specimens, with the 0.3 mm layer thickness showing the lowest roughness. They also revealed that mechanical finishing with a series of SiC sandpapers of increasing grit (P80, P180, P320, P800, P1200, and P4000) effectively reduced the specimens' surface roughness.

Future research, with larger samples, should investigate how to optimize the printing process to minimize the roughness of the printed specimens and, consequently, decrease the need for mechanical surface finishing after printing.

**Author Contributions:** Conceptualization, T.S., C.F. and C.B. methodology, C.F. and C.B.; software, C.F.; validation, C.B. and M.H.F.; formal analysis, M.H.F., C.B. and T.R.; investigation, T.S.; resources, C.F. and M.A.P.V.; data curation, T.S., C.F., C.B. and T.R.; writing—T.S. and T.R.; writing—review and editing, T.S., M.H.F., C.B. and M.A.P.V.; visualization, M.H.F., C.B. and M.A.P.V.; supervision, M.A.P.V.; project administration, T.R.; funding acquisition, C.F. and M.A.P.V. All authors have read and agreed to the published version of the manuscript.

**Funding:** This research received no external funding.

**Data Availability Statement:** Data are contained within the article.

**Acknowledgments:** The authors thank those who voluntarily helped in the 3D printing and polishing of the specimens, namely, Bruna Oliveira of INEGI's Product Development Engineer Department and Emília Soares of the FEUP's materiology laboratory. Carlos Fernandes is grateful for the funding through LAETA, Portugal, in the framework of project UID/50022/2020.

**Conflicts of Interest:** The authors declare no conflicts of interest.

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
