# Peer review of "Surface Roughness of Polyetheretherketone Printed by Fused Deposition Modeling: A Pilot Study Investigating the Impact of Print Layer Thickness and Polishing Method"

_applsci, doi:10.3390/app14073096_

Round 1

Reviewer 1 Report

Comments and Suggestions for Authors

The article is well written and is nice to read. I don't see any reasons why it should not get published. I do have just a bunch of minor revisions to suggest.

1. Figure 5. First of all, the letters (a), (b), (c), (d), (e), (f) and (g) are not clearly readble. Please add them, even if it is quite clear which is which. Moreover, I believe it would help readbility if figs (d), (e), (f) and (g) had the same color scale, as it would make it easier to compare them.

2. The results that are obtained are somehow surprising, as one would expect a lower roughness with the 0.1 mm layer thickness. The authors do provide a sort of explanation, but i did not find it very clear. I ask the authors to give a clearer explanation.

3. The authors correctly point out that the results are obtained on a minimal number of specimens, and this is the main limitation. Although I do appreciate this comment, I ask the authors, would it be so difficult to perform testing on a few more specimens?

Reviewer 2 Report

Comments and Suggestions for Authors

The manuscript deals with interesting topic. The study is within scope of the journal. There is evident certain level of innovativeness. The methodology is reasonable and well described. Obtained results are interesting and applicable. However, there are few issues which has to be addressed prior the acceptance of this paper for publication:

1. Abstract part: there is no need to introduce abbreviations here. Anyway, all abbreviations are introducen in an Intriduction part.

2. line 133: alphabet x versus symbol ×: I recommend to use symbol × . It means " (10mm x 10mm x 1.5mm)" has to be rewritten to the form " (10mm × 10mm × 1.5mm)".

3. line 170: the same problem.  (10 x 10 x 1.5mm) to  (10 × 10 × 1.5mm) 

4. Table 5: error analysis is missing. At least standard deviation of the values has to be mentioned.

All in all, the manuscript is interesting. However, some issues have to be addressed.

Reviewer 3 Report

Comments and Suggestions for Authors

1. For the values in Table 1 ~Table 5: I think that it is appreciated to use the ‘period’ (‘.’) instead of the ‘comma’ (‘,’).

2. What is the difference between A1 and A2 samples, B1 and B2 samples? Please describe clearly in the ‘2. Materials and Methods’ section.

3. In the caption of Figure 5: (a), (b), (c), (d), (e), (f), and (g) mentioned by Auhtors do not match the images. Please make it clear.

4. In this work, why does the sample with 0.3 mm of print layer thickness have a smoother surface than the sample with 0.1 mm after polishing?

5. How does the smoothness of the sample surface change after polishing if the print layer thickness is thinner or thicker?

---------------------------------------------The end------------------------------------------

Comments on the Quality of English Language

Overall, the manuscript is considered to be well-written in English. However, please check again for typos or awkward expressions.

Round 2

Reviewer 3 Report

Comments and Suggestions for Authors

The authors' responses to the reviewer's questions were thought to be clear and conscientious. And it seems to have been edited to reflect those contents well in the text. Therefore, this manuscript is considered acceptable in this journal.